# Transglutaminase in Foods and Biotechnology

**DOI:** 10.3390/ijms241512402

**Published:** 2023-08-03

**Authors:** Katja Vasić, Željko Knez, Maja Leitgeb

**Affiliations:** 1Laboratory for Separation Processes and Product Design, Faculty of Chemistry and Chemical Engineering, University of Maribor, Smetanova Ulica 17, SI-2000 Maribor, Slovenia; katja.vasic@um.si (K.V.); zeljko.knez@um.si (Ž.K.); 2Faculty of Medicine, University of Maribor, Taborska Ulica 8, SI-2000 Maribor, Slovenia

**Keywords:** transglutaminase, industrial enzyme, protein-modifying enzyme, crosslinker, antibody–drug conjugation, immobilization

## Abstract

Stabilization and reusability of enzyme transglutaminase (TGM) are important goals for the enzymatic process since immobilizing TGM plays an important role in different technologies and industries. TGM can be used in many applications. In the food industry, it plays a role as a protein-modifying enzyme, while, in biotechnology and pharmaceutical applications, it is used in mediated bioconjugation due to its extraordinary crosslinking ability. TGMs (EC 2.3.2.13) are enzymes that catalyze the formation of a covalent bond between a free amino group of protein-bound or peptide-bound lysine, which acts as an acyl acceptor, and the γ-carboxamide group of protein-bound or peptide-bound glutamine, which acts as an acyl donor. This results in the modification of proteins through either intramolecular or intermolecular crosslinking, which improves the use of the respective proteins significantly.

## 1. Introduction

Enzymes were used long before the development of modern DNA technology as fermenting microorganisms or crude preparations of different fruits. However, with the development of advanced bioprocesses using recombinant DNA technology, enzymes are being purified and produced on a larger scale, which has allowed their use and their applications in different industrial technologies, such as in the chemical, food, textile, cosmetic, pulp, and paper industries [1,2,3,4]. They are nutrients that play an important role in the physical properties of different foods. As such, they have been used widely in many industrial processes for the production of different products. Concerns regarding global food shortages and population growth, as well as the use of advanced food proteins, are increasing constantly in recent years. Moreover, the importance of protein modification technology is gaining much interest in order to meet the ever-challenging needs of a growing population [1,5,6,7].

The development of protein engineering with site-directed evolution has enabled novel enzymes with enhanced activities for many new processes, which makes industrial enzymes that are needed in everyday life more accessible to various industries [8,9]. Among the most-used industrial enzymes are hydrolases and carbohydrases. Hydrolases, such as lipases and proteases, are the dominant type used in many food industries, mostly dairy, as well as in the detergent and chemical industries. Carbohydrases include amylases and cellulases and are also used extensively. Table 1 lists some industrial enzymes with their significant industrial applications.

Drastic savings in resources were achieved by applying industrial enzymes in various processes; for example, energy efficiency and water and raw material consumption have improved significantly. Among other approaches, when compared to chemical modifications, applying enzymes in protein modification displays many advantages, which include high reaction specificities and low side-reaction frequencies, with the lack of need for high-pressure and high-temperature conditions. Such advantages make the protein modification technology effective, mostly in the food industry [10,11,12,13,14]. In terms of applying hydrolases in industry, proteases were the main protein-modifying enzymes. With the emergence of the enzyme TGM, which is involved in protein crosslinking, the protein modification technology possibilities have expanded enormously [15,16,17,18]. TGM (EC 2.3.2.13) belongs to the transferase family, distributed widely in nature. TGM is responsible for acyl transfer, deamidation, and crosslinking of intra- or inter-chain glutamine peptide moiety, which is the acyl donor and lysine peptide moiety, which is the acyl acceptor. The enzyme TGM catalyzes the addition of free amines into proteins by joining the glutamine residue. When the amine is absent, water becomes the acyl acceptor, and the γ-carboxamide groups deamidate to glutamic acid residues. The transamidation reaction occurs when the ε-amino groups of lysine residues in proteins act as acyl acceptors. In such cases, the acyl transfer onto a lysine residue forms intra-molecular and inter-molecular covalent crosslinks of ε-(γ-glutamyl)lysine, which is enriched with essential amino acids [19,20].

TGMs can be found in plants, such as soy, topinambour, and fodder beet; in animals, such as animal body fluids and fish; as well as in microorganisms. TGMs from mammalian sources are Ca^2+^-dependent, while microbial TGMs are Ca^2+^-independent and have smaller MW. Due to their Ca^2+^ independency, microbial TGMs are considered to be more cost-effective and eco-friendly, while their characteristics can prevent changes in formation of by-products, which occurs in Ca^2+^ protein complexes [21,22]. Moreover, their source origin also dictates their activities, which varies depending on their origin. The main differences between microbial TGM and existing TGMs from animal sources are presented in Table 2.

TGM-catalyzed reactions result in functional property changes, such as solubility foaming, viscosity, elasticity, water holding capacity, emulsifying capacity, gelation, and thermal stability of different food proteins [23,24,25]. TGM was also found to be involved in many physiological processes, such as in coagulation antibacterial immune reactions and in photosynthesis. TGM is an extracellular enzyme and was isolated from *Streptoverticillium* sp. And *Physarum polycephalum*. It can also be biosynthesized by many microorganisms, such as *Streptoverticillium* sp., *Streptoverticillium cinnamoneum*, *Streptomyces netropsis*, *Streptoverticillium ladakanum*, *Streptomyces lydicus*, and *Bacillus subtilis.*

## 2. Enzymatic Properties of TGMs

TGM modifies proteins with amine incorporation and crosslinking, where TGM catalyzes the reaction of the acyl transfer between the γ-carboxyamide group of peptides, which is bound with glutamine residue acyl donors and primary amines receptors of different compounds. The reaction can be found in Figure 1a. As presented in Figure 1b, the ε-amino group of lysine reacts as a receptor, which forms intra-molecular and inter-molecular crosslinks of ε-(γ-glutamyl)lysine isopeptides. When the lysine residue is absent, or when the protein system is free, water reacts as the receptor for the acyl groups, and the carboxyamide groups of the glutamine residues are deamidated, which, consequently, forms glutamic acid and ammonia residues that can modify protein charges and protein stability. The reaction is presented in Figure 1c.

The TGM crosslinking reaction occurs before the acyl transfer and deamidation reactions in food systems, which results in the formation of glutamyl lysine isopeptides and polymers with high molecular weight. Consequently, it changes the functional properties of the proteins, resulting in improved rheology and other quality properties for various food products (Figure 2) [26]. The properties and structure of TGM were studied by many researchers, where it was reported that the MW of microbial TGM is approximately 38,000 kDa and is only half of its MW when originating from an animal source [27,28]. The microbial TGM consists of 331 amino acids in only one polypeptide chain. The secondary structure of TGM consists of eight β-strands, which are surrounded by eleven α-helixes. Different 3D structures of TGM from different sources are presented in Figure 3. Microbial TGM has a stabile catalytic activity over a wider range of pH values in comparison to animal TGM. The optimal pH activity of microbial TGM is in the range 5–7 and has the isoelectric point of 8, while the optimal catalytic activity of TGM lies in the temperature range from 40–50 °C, at pH 6. When the temperature increases over 70 °C, the activity decreases drastically, resulting in activity loss [26,29,30].

The protein crosslinking reaction, called polymerization, can result in dimer, trimer, and polymer formation. For identifying these crosslinked formations, commonly used techniques are gel electrophoresis (GE), size exclusion chromatography (SEC), and isopeptide content quantification (ICQ). GE allows identification of casein molecules’ crosslinks, such as aS1-, aS2-, and κ-casein, whereas high-MW polymers with over 250 kDa cannot go through the gel [31]. The quantification of monomer conversions to dimer, trimer, and oligomer is identified by SEC, which provides an estimation of the polymerization degree (PD), also considered the crosslinking degree. The PD is the ratio between the dimer, trimer, and oligomer sum and the monomer, dimer, trimer, and oligomer sum. The ICQ technique can also quantify the crosslinking reaction since the isopeptide bonds form during the crosslinking reaction and do not succumb to protein hydrolysis [32,33].
Figure 3TGMs and their 3D structures from animal source (*Pagros major*), microbial source (*Streptomyces mobaraensis*), and plant source (*Phytophthora sojae*) (RCSB Protein Data Bank [34]).
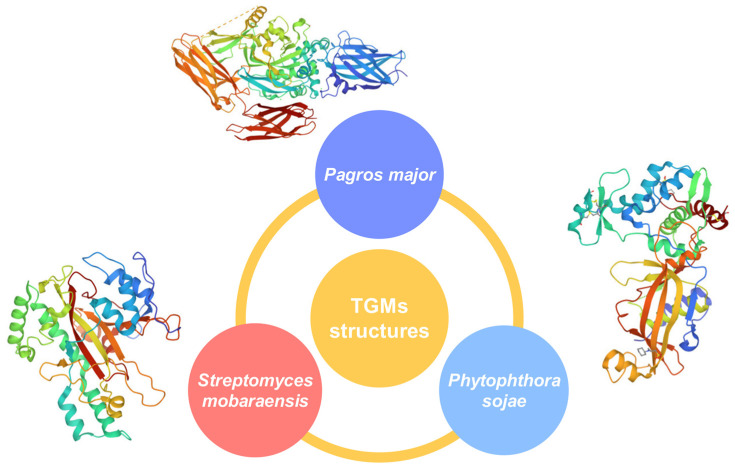



## 3. Origins of TGMs

Based on the similarities of catalytic properties and mechanism of TGM reactions, TGMs are believed to be evolutionary, close to thiol-like proteases. Clusters of TGM-like domains can be identified in eucaryotic TGMs that are found in all archaea and some yeast and bacterial species [35,36,37]. TGMs are enzymes found both on the inside and outside of the cell, which determines the versatility of their functions. Enzymatic activity of TGM was found in microorganisms, plant, and animal tissues. They all exhibit similar catalytic activity with biochemical properties, although those from plant and animal sources have less homology in the composition of amino acids [38,39,40]. TGMs from animal sources take part in many physiological processes, such as participating in skin formation, blood coagulation, and in antimicrobial immune reactions. Additionally, TGMs from plant sources play a role in the process of growth and development [41,42]. A specific feature of TGMs from plant sources is their sensitivity to light. This property applies especially to chloroplast TGM, which has been investigated by many studies [43,44]. Isolation and purification of TGM from microbiological sources has allowed its application in many processes and its simplification, which has provided economical savings with lower energy consumption. Gene transfer technology provided many possible TGM productions, where the expression of genes in *E. coli* has increased the production of TGMs and their efficiency immensely. Such enzymes are consumer-friendly and biodegradable, which offers a great advantage over many other chemical substances [45,46,47]. TGM is well-known for intra-molecular and inter-molecular formations of covalent bonds of glutamine and lysine, which initiate high-MW peptides, such as monomers, dimers, trimers, and oligomers. The digestibility of such crosslinked peptides has raised nutritional concerns. After ingesting crosslinked peptides, the dipeptide (glutamine–lysine) is cleaved by the activity of two enzymes. γ-glutamylamine cyclotransferase is a kidney enzyme cleaving the glutamine–lysine isopeptide, which yields free lysine and 5-oxo-proline, and is later metabolized to glutamate by 5-oxo-prolinase. γ-glutamine transpeptidase can be found in the intestinal membrane, kidneys, and blood. Microbial TGM is cultivated from *Streptoverticillium* strains [48,49,50,51]. However, the large amount of highly expensive nutrients makes its production not attractive from the economical angle. Many researchers have studied the use of agricultural waste materials as a source of carbon for TGM production. The media used to produce microbial TGM contain yeast extract, peptone, potassium and sodium phosphate, magnesium sulfate, and a source of carbon. The source of carbon may be xylose, which is a hemicellulose sugar used for bacterial proliferation [52]. The biosynthesis of TGM can be performed on different batch cultures, where the medium contains saccharose, glucose, starch, or dextrins as a carbon source. The culture *Streptovitricillium mobaraense* was found to be the most suitable medium, which contained corn steep, aminobac, and yeast extract as a source of nitrogen [53,54,55,56]. However, peptone, yeast extract, urea, and casein are used commonly as sources of nitrogen that are used in TGM biosynthesis. However, nowadays, TGM is usually produced by *Streptovitricillium mobaraense* in fermentation systems, which are followed by the downstream process presented in Figure 4. The main advantages of TGM production from microorganisms, compared to production from animal and plant sources, are the purity and high productivity since there is no need for difficult separation and filtration processes.

As eucaryotic TGMs are found in phylogenetic taxonomic groups, such as plants, animals, and fungi, the procaryotic TGMs are found in microorganisms. TGM was first isolated from *S. mobaraensis* in 1989, and later found in many other strains, listed in Table 3.

Microbial TGM from *S. mobaraensis* is used widely in different food industries, where it is applied in the reconstruction and manufacturing of meat, texturization of dairy products (yoghurt [42,89,90,91,92] and cheese [42,93,94,95,96]), or in different materials sciences, where it is used for the stabilization of wool and leather [97,98]. Due to the various applications of TGM, different grades of enzyme are available commercially. For biotechnological technologies, highly pure enzyme TGM is recommended to surpass by-products that can appear during side-reactions.

## 4. Applications of TGM

Enzyme TGM gained much attention due mainly to its potential for industrial applications. In their earlier uses, TGMs focused mostly on the cheesemaking processes, mainly improving the characteristics of the cheese itself. Later on, beverage properties were being improved by decreasing viscosity and solubility increases, as well as reduction in yoghurt syneresis. Therefore, the majority of investigations of applying TGMs in different processes focused mostly on improving the functional properties of proteins that can be utilized to develop improved food ingredients, such as crosslinked milk powders and high-added food products [33]. Such improvements were also possible due to the different immobilization techniques of TGM.

### 4.1. TGM Immobilization

The immobilization of TGM on solid supports is a widely used method to increase TGM stability and improve its spectrum of use and reuse. Site-specific modification is needed for such applications, especially when unstable targets are in question. Harsh conditions are usually applied during chemical immobilization. If compared to enzymatic catalysis, selective and fast performance in aqueous media is performed under mild process conditions, which has many advantages. Immobilizing the enzyme onto a given matrix is benefiting the enzyme with a stabile structure, which provides an advantage in resisting temperature and pH alterations by retaining catalytic activities [99,100,101,102]. Immobilized enzymes are more stable and have improved properties and features in terms of their kinetic aspects when compared to the free form of the enzyme. The enzymes’ improved features are due to the conformational changes that happen in the enzyme structure as a result of the chosen and most suitable immobilization method. In this manner, enhanced activity, stability, and selectivity can be observed [103,104,105,106]. For larger-scale applications, immobilized enzymes are the considered subject of choice. Not many studies were reported on immobilizing TGM. However, immobilized TGM was investigated on a few supports, such as thermo-responsive carboxylated poly (N-isopolylacrylamide), agarose beads, polypropylene microporous membranes, and various nanomaterials, such as magnetic nanoparticles (MNPs) or carbon nanotubes (CNTs), as well as in the form of crosslinked enzymes aggregates (CLEAs). For example, TGM was immobilized on multi-walled CNTs for tissue scaffold designing. The highest immobilization efficiency of 58% was achieved and a 4.8 fold increase in catalytic efficiency was observed [107]. Gianetto et al. reported on a new amperometric immunosensor, which was based on the covalent immobilization of TGM onto functionalized gold nanoparticles, and used for the determination of anti-tissue TGM antibodies in human serum [108]. Another piezoelectric immunosensor was developed for the detection of anti-tissue TGM antibodies as specific biomarkers for early diagnosis of celiac disease by Manfredi et al. [109]. Leitgeb et al. studied and investigated the immobilization of TGM onto surface-modified MNPs with carboxymethyl dextran for cleaner production technologies. In this case, the TGM was hyperactivated and exhibited 99% immobilization efficiency with 110% residual activity. It showed excellent thermal stability at 50 °C and at 70 °C [110]. Another TGM investigation by the same group reported on the synthesis of TGM immobilized in the form of CLEAs and magnetic CLEAs (mCLEAs). The TGM was precipitated in 2-propanol and later crosslinked with glutaraldehyde (GA), which resulted in 63% and 73% of residual activity for TGM CLEAs and mCLEAs, respectively. The CLEAs and mCLEAs showed great immobilization efficiency as well (95% and 90% for CLEAs and mCLEAs, respectively) [111]. Zhou et al. reported on TGM immobilized covalently on thermo-responsive carboxylated poly(N-isopropylacrylamide), where the immobilized TGM exhibited reversible solubility in an aqueous solution with a low critical solution temperature of 39 °C. Such immobilized TGM can be used to modify proteins in food processing and biomedical engineering [112]. The Wen-qiong report showed immobilization of TGM on an ultrafiltration polyethersulfone membrane surface, where it retained 50% of residual activity after 20 days. Additionally, the TGM-immobilized membrane had a higher relative membrane flux of 0.15 MPa in a membrane reactor [113].

### 4.2. Food Related Industries

As proteins are important food components, which play an important role in the phyisicochemical properties of food, the usage of TGM to crosslink food proteins to change their functional characteristics has been in progress for more than 30 years [114,115,116]. Enzymatic preparations of TGM have an important role in the food industry due to their practical utilization. Many reports describe the use of TGM in various food-related industries for the crosslinking of proteins, as in meat, cheese, yogurt, or bread (Figure 5).

It can also be used to produce composite edible films. TGM catalyzes the formation of crosslinks within a molecule as well as between molecules of other proteins. This feature impacts the changes in protein functionalities, such as solubility, foaming, emulsifying capacity, and gelation. As TGM has broad substrate specificity, Table 4 shows the reactivity of microbial TGM, which was investigated on different types of proteins that were derived from various foods [117,118,119,120].

#### 4.2.1. Dairy Industry

TGM uses proteins such as casein or whey proteins as substrates to improve the foaming, emulsifying, and gelling properties of different foods. Casein is a major milk protein, which is a great substrate for TGM in dairy products due to its low degree of tertiary structure, flexibility, and the absence of disulfide bonds, which allows the exposure of reactive groups to TGM. On the other hand, the globular whey proteins, which do contain disulfide bonds, are poor substrates for TGM in the crosslinking process and therefore require modifications. Such modifications can be performed with reducing agents or increasing the pH value. They can also be achieved by heat denaturation or application of high hydrostatic pressure. However, such alterations in treatment can affect the interactions between the enzyme TGM in the TGM inhibitors that are present in milk serum and can also induce denaturation and result in cleavage of disulfide bonds, which later leads to unfolding of the proteins [26]. For example, the enzymatic crosslinking of casein was more resistant to digestion in comparison to the non-crosslinked casein. This suggests that the development of new types of products can offer carious food with improved structural characteristics, such as the polymerization of milk proteins with TGM results in protein film formation, which improves the functional properties of dairy products [121,122].

In the dairy industry, TGM has been introduced in many products. In yoghurts, it is used to prevent syneresis and for texture firming or softening since TGM-modified casein allows the manufacture of dairy products with a more consistent structure. The result of this reduced syneresis is a firmer and more homogeneous product. Various methods and protocols were also carried out investigating the use of TGM to increase cheese yield while enhancing the quality of low-fat cheese. TGM is also used in cheese manufacturing, where three methods are performed, including TGM:-the addition of TGM to milk, followed by heating for pasteurization and deactivation of enzymes, concluded with the addition of rennet to the milk;-the addition of rennet to the milk, followed by the addition of TGM;-the addition of TGM and rennet at the same time.

However, the reported investigations confirmed that the addition of TGM before the rennet prevented the coagulation of milk, while the simultaneous addition of both resulted in reduced resistance and hardness of the cheese [123]. By improving the cheese yield, textural properties, and its water-holding capacity, the use of TGM in cheese production is crucial. Other investigations reported on the improved heat stability and consistency of processed cheese after implementing TGM into the production [124,125,126,127,128].

Microbial TGM was used to treat the rheological and microstructural properties of yoghurt, where it was applied to milk before the fermentation. The TGM-mediated treatment decreased the ropiness of yoghurts and contributed to the acceptability of their texture, a study by Marhons suggests [129]. Salunke et al. investigated the use of micellar casein concentrate and milk proteins that were treated with TGM in different imitation cheese products. As TGM has the potential to modify the surface properties of milk protein concentrate and micellar casein concentrate, it may also improve functionality in imitation cheese, such as mozzarella [130]. Another study by the same group investigated the melt and stretch properties of dairy-based imitation mozzarella cheese, where the effect was studied of TGM-treated concentrates. The results demonstrated that TGM treatment modifies the investigated stretch and melt functionalities of milk protein concentrate and micellar casein concentrate [95]. Another study by Monsalve-Atencio et al. investigated the effect of TGM and its interaction with another enzyme, phospholipase, on the composition, yield, texture, and microstructure of semi-soft fresh cheese. The interaction of TGM with phospholipase showed the highest content of moisture in cheese value, which suggests an economically improved application of TGM in cheesemaking [131]. The addition of microbial TGM in quark cheese was studied, where the physicochemical, textural, sensory, and microbial properties of cheese were studied as well [132].

#### 4.2.2. Baking Industry

The use of TGM in the baking industry is improving the quality of flour, and, consequently, the texture and volume of bread as proteins from grains are good substrates for crosslinkers by TGM. For example, rice flour is known to contain valuable nutrients, such as proteins, vitamins B and E, as well as fiber. However, it can only be used in and is limited to non-fermented bakery products. Investigations showed that the addition of TGM to rice flour improved the rheological properties of dough, and, by that, increased the content of triglycerides [133,134,135,136]. Similar studies were reported concerning cassava and wheat flour [137,138]. A TGM-induced protein aggregation method to improve the baking properties was investigated by Beck et al. [139], where the effect was studied of microbial TGM on the properties of rye dough. It was reported that the addition of TGM modified the rheological properties of rye flour dough, which resulted in a progressive increase in shear modulus. The increased TG concentration also showed an increase in crumb springiness and hardness, which demonstrated the improved breadmaking with the use of TGM. Another study also reports on improved rheological properties of gluten-free batter with the implementation of TGM. The crumb properties revealed that increased TGM concentration increased crumb chewiness and firmness [140]. The use of varying amounts of TGM also improved the baking quality of high-level sun pest wheat, where it was observed that TGM plays an important role in the baking quality. Increasing TGM activity caused increased bread characteristics of wheat, such as bread yield, height, crumb softness, pore structure, as well as decreased weight loss and wideness of the bread samples. The study concluded that the addition of TGM can restore the properties of bread and improve its overall protein structure [141]. Lang et al. evaluated the influence of TGM on the technological properties of gluten-free cakes of brown, black, and red rice. The effect of baking on the phenolic compound content was investigated as well [135].

#### 4.2.3. Meat Industry

Numerous reports and studies are available investigating the use of TGM in meat products as one of the most widespread applications of TGM is in the restructuring of meat. Despite improving the structure and texture of the meat product, the use of TGM also provides cohesion without thermal processing or any additives, such as phosphates [59,142,143,144,145]. Studies show that the crosslinking activity of TGM in meat depends strongly on the temperature, pH, protein surface charge, and ionic strength. It was shown to improve other characteristics as well, such as water-binding, gelation, and emulsion stability. With the use of TGM in meat production, the secondary structure of a myosin heavy chain is changed by reduced α-helix content and increasing β-sheet content, which results in the formation of high-molecular-weight polymers. With such structural modifications, strong gels were formed with compact structural properties, which allow cohesiveness and improve the hardness of the meat. In addition, some studies also show that different degrees of gelation can be observed when induced by the addition of TGM [146,147,148]. This is also valid for fish skin gelatin. Namely, TGM-modified cold-water fish skin gelatin could be a potential mammalian gelatin replacer [149]. The addition of TGM also improves the quality of collagen, blood proteins, and provides higher nutritional value by supplementing with respective amino acids, such as exogenous lysine. TGM has allowed the production of new meat products, which use lower-quality raw materials instead of high-value meat products. The impact of TGM on the protein of such raw materials (skimmed milk powder or soy powder) does not alter the appearance, smell, texture, or nutritional value from products made with high-quality meat [150]. A study by Ribeiro et al. produced bovine meat with different levels of TGM combined with papain. The effects were investigated on pH, water activity, instrumental color, proximate composition, texture, and yield. It was concluded that the addition of TGM increased the yield of meat loafs [147]. In an interesting study by Wen et al., the enzyme TGM was used to develop 3D-printable meat analogs that imitate the physiochemical properties of beef. The TGM improved the rheological properties of raw meat, and provided a method for modifying the texture of meat analogs using TGM catalysis [151].

As the majority of the population are omnivores, TGM is being used widely in plant-based (PB) food industries, which are designing PB products that mimic the look, taste, and feel of animal-sourced foods. Zhou et al. developed PB protein gels for meat analogues that are created using slats, polysaccharides, and crosslinking enzymes, such as TGM, to modulate their gelation and assembly properties. In their study, the TGM increased the gel strength by forming covalent crosslinks between the potato protein molecules with more meat-like structures [152]. Additionally, traditional sausage production technologies can be used for PB analogues, such as a PB salami-type sausage analogue, which was manufactured with TGM-mediated soy protein isolate gels as binders, investigated in a study by Herz et al. [153].

In addition, the use of TGM is gaining much attention in the field of food packaging products; e.g., hemp proteins were used as raw material to obtain biodegradable films since they were demonstrated to act as both acyl donor and acceptor substrates of microbial TGM crosslinking [154,155]. Because such bioplastics show higher gas permeability and greater hydrophobicity, they may be useful as packaging systems for protecting food products from physical contamination and, thus, for extending their shelf-life. In the crosslinked gelatin-based films, food preservatives such as lysozyme or nisin may be incorporated to extend the shelf life of perishable foods. It was demonstrated that microbial-TGM-crosslinked gelatin-based films incorporated with lysozyme can control the release of this food preservative effectively.

### 4.3. Biotechnology and Cosmetics

Microbial TGM is an interesting tool for protein modification, which catalyzes protein crosslinking through isopeptide bond formation, which occurs between γ-carboxamide groups of glutamine residues, which include the acyl donor, and primary amines, such as ε-amino groups of lysine residues, which include the acyl acceptor. Therefore, research on TGM use can be applied to biomedical, biomaterial, cosmetic, and feedstock technologies. TGM can be used for modification of gelatin hydrogels and collagen for enhancing binding in different tissues. Microbial TGM was used to prepare collagen-grafted chitosan, which could serve not only to reduce the loss of moisture but also to absorb the moisture. With such properties, it showed the potentiality to repair skin in the cosmetic, biomedical, and pharmaceutical fields [156]. TGM-crosslinked whey proteins were also used to prepare a D-limonene emulsion, which can solve the problems of easy oxidation and poor water solubility of D-limonene. Limonene is an important ingredient in the formulation of different cosmetic and personal care products, such as aftershave lotions, bath products, cleansing products, eye shadows, hair products, lipsticks, shampoos, etc. [157]. Moreover, microbial TGM, as a protein-crosslinking enzyme in the processing of hair, improved the rigidity of hair fibers by 15.64% compared to a control when it was applied to damaged hair [158].

It has also been applied for bioconjugation, in order to create antibody–drug conjugates for various therapeutic applications. Additionally, new uses for TGM in the field of novel biomaterials are suggested and can be generated via site-specific substrate binding by proteins modified by TGM [159,160]. Regarding different feedstocks, TGM has shown to improve the physical properties of fish feed, while, in cosmetics, the TGM-catalyzed reactions between amino groups of starch and γ-carboxamide groups of collagen peptides were investigated and reported to increase the effectiveness of some synthesized materials, such as drugs [156,161,162].

Fusion proteins with dual functions are important for immunochemical assays. Among such assays are the enzyme-linked immunosorbent assay (ELISA) and Western blot assays. In that manner, genetic fusion provides poor yields when large-sized hybrid molecules are assembled. Therefore, TGM catalysis is considered as a method for the preparation of protein–protein conjugates. It was demonstrated that the coupling of two functional proteins, namely peroxidase and protein G, is possible through lysine and glutamine active sites. As a result, only a small amount of the desired conjugate was yielded. Later, when TGM-mediated conjugation was performed, only the desired conjugates were obtained [163,164]. This finding provided TGM the recognition that respective tags can be applied to recombinant production to terminal and internal sites. Therefore, acyl donor and acceptor incorporation enable covalent linkage in the monomeric subunits, which enhances the thermal stability of the dimer [165]. Native antibody site-specific modification enhances the properties of antibody-based bioconjugates. However, such antibodies have a single functionality. A work by Walker et al. addressed this limitation by designing heterofunctional substrates for microbial TGM that can contain both azide and methyltetrazine “click handles”, which present a powerful method in the toolbox for native antibody modification [166,167,168,169,170]. Antibody–drug conjugates for cancer treatment have placed site-specific TGM-catalyzed conjugation with cytotoxic properties at the very pinnacle of research. TGM’s remarkable properties make it a versatile tool for post-translational modification of various proteins (Figure 6). A few examples are PEGylation of small-protein drugs to elevate their half-life, or immobilization of biocatalysts that are prone to aggregation in order to increase their stability or covalent attachment of nucleic acids to proteins for combining the properties of both biomolecules [171,172,173,174]. The reactivity of microbial TGM was investigated at intrinsic lysine and glutamine sites of different antibodies [167,170,175,176,177,178]. The amino component in N-terminal pentaglycyl was found to mediate protein modification by TGM, which was performed by crosslinking of the enhanced green fluorescent protein. Many other protein–protein conjugates were performed using TGM-catalyzed conjugation. As reported by Bhokisham et al., attachment on solid supports occurred, which caused optimization of the reaction stoichiometry [179].

Covalent coupling of nucleic acid macromolecules, such as DNA and RNA, to proteins is a powerful method in molecular biology since both components exhibit excellent functions [180,181,182]. Therefore, protein–oligonucleotide conjugates became valuable tools in analytic and biomedical applications, for example, in drug delivery [183] and molecular diagnostics [184].

Protein–oligonucleotide ligation often relies on chemical methods that include functional groups of the desired target protein. As a result, partial functional loss may occur due to the steric hindrance or protein treatment with organic solvents. Therefore, site-specific coupling is performed to avoid residue modification, which is essential for protein function. In that manner, innovative strategies were developed to include microbial TGM in the modification process. Some research reports show aminated DNA, which was attached chemically to the acyl donor substrate and further coupled to alkaline phosphatase [185].

A modern technique to improve the pharmacokinetic properties of pharmacophore is PEGylation, where polyethylene glycol is attached to small-protein-based drugs. PEGylated pharmaceutics usually reduce immunogenicity compared to the non-PEGylated counterparts due to the enlarged hydrodynamic radius and higher conformational stability [186]. With increased bioavailability, the intravenous administration of such protein-based drugs can become more patient-friendly. The controlled conjugation strategies are very important since the modifications at binding interfaces or active sites of residues can affect the in vivo functions. Leading to undesired reactivity during conjugation, it can lead to a heterogeneous mixture of products with changed pharmacokinetic properties. Nevertheless, PEGylated proteins that are synthesized by chemical derivatization are already available on the market [186]. However, compared to chemical modification of such proteins, the site selectivity of microbial TGM is accessing PEGylated derivatives of various drugs without altering their properties [187].

Another way to manipulate specificity is covalent immobilization of TGM on solid supports. In a study by Grigoletto et al., TGM was coupled to agarose beads through an N-terminus to investigate its activity and substrate specificity. The immobilized TGM exhibited changed enzymatic activity and kinetic parameters, which were the result of chemical modification. Due to PEGylation, the immobilized TGM appeared more site-selective [171]. A biodegradable alternative to PEG, hydroxyethyl starch, is a polymeric molecule, which was used for microbial TGM-mediated protein conjugation and served as an acyl donor/acceptor to ligate monodansyl cadaverine [188]. Moreover, microbial TGM was also proved to be able to catalyze an acyltransfer reaction between the aminated oligosaccharides and acyl donor molecules [189], while glycosylation of catalase and trypsin was obtained by the transamidation of carboxamide functions.

Antibody–drug conjugates are promising tools for tumor treatments, where a chemotherapeutic is bound covalently to immunoglobulin, with the intent to enlarge the therapeutic range with the combination of powerful organic toxins and targeted specificity antibodies. Cytotoxic drugs are used widely to treat malignancies and solid tumors, and have, under specific clinical conditions, altered the natural course of certain diseases. Due to their intrinsic mode of active site, they are effective but can also cause significant on-target events that could result in the discontinuation of medication, which would increase the risk of recurrence of the tumor. For maintaining the efficiency of chemotherapeutics, efforts were undertaken to investigate novel approaches. Among such approaches was the conjugation of cytotoxic agents to antibodies. Many studies describe improved pharmacokinetics, enhanced efficacy, and reduced toxicity of antibody–drug conjugates when they are ligated site-selectively [190]. Non-directional methods generate heterogeneous products with statistically distributed coupling sites and fluctuating hydrophobic profiles, while site-specific conjugation provides reproducible hydrophobic properties with antibody modification restrictions (Figure 7).

Most of the antibody–drug conjugates in clinical use nowadays are assembled by random modification of cysteines and lysines. Many studies where an enzymatic approach was developed to modify antibodies site-selectively were reported [191,192,193,194,195,196]. Such an approach includes additional cysteine or selenocysteine residues [197,198]. Microbial TGM shows incapability of labelling glutamines in native human antibodies efficiently, although many Gln sites are exposed and available [199]. Therefore, a specific recognition tag is being incorporated and is reported to be a powerful strategy in overcoming such limitations. Research was performed at Pfizer, where a TGM recognition sequence was placed at various surface-exposed regions of a specific growth factor receptor antibody, which facilitated efficient labeling by microbial TGM [165].

In the synthesis process of an antibody–drug conjugate, all aspects must be taken into consideration. The target, antigen, antibody, linker, and the cytotoxic load must be evaluated for the targeted cancer indication. The antibodies with cytotoxic load must obtain excellent targeting capabilities to distinguish between healthy and tumor cells. In the process, antibodies that are engineered to follow a specific tumor antigen attach themselves to the surface of tumor cells. When processing within lysosomes or endosomes takes place, the antibody–drug conjugate releases its lethal load and destroys the targeted tumor cells (Figure 8). Due to its highly targeted tumor antigen expression, recognition, and requirements for effective internalization and processing, the antibody–drug conjugates are believed to provide a broader therapeutic opportunity in the ever-growing field of enzyme-mediated modification strategies [168].

## 5. Conclusions

TGMs remain one of the most complex families of enzymes that possess various structural and functional properties in mammalians, non-mammalian eucaryotes, and in bacteria. As highly efficient enzymes with unique features, they are used in the development and improvement of many versatile products. The extraordinary applicability of TGMs in the formulation of different dairy, meat, and feedstock products has led to an incredible enhancement in the food industries, where they have been proven to be efficient and applicable tools for the development of new products. Aside from the extensive use of TGM in food-related and manufacturing industries, multiple achievements have contributed to biotechnological research, where it is used as an antibody–drug conjugate, and such research has become a promising field for its refinement. Site-specific conjugation is a widely developed strategy to customize the properties of target proteins towards various applications in pharmaceutical and biomedical applications, by loading tumor-specific antibodies with small cytotoxic molecules, in order to create antibody–drug conjugates. In recent years, numerous studies have reported on the considerable impact that enzyme TGM has in either immobilized or free form. For example, while compared to traditional conjugation methodologies, TGM-mediated catalysis has many beneficial advantages. TGM is a powerful component in the biotechnological field, and an important tool in the bioconjugation process. Furthermore, TGM has the potential to be useful in many non-food-related fields as well, where it will continue to contribute to new innovative products, including using novel technologies for enzyme modifications by protein engineering and for cleaner productions. Reducing the costs of production is essential and is aiming and guiding their applications on a larger scale in various industrial sectors, such as in the design of different biotechnological products.

## Figures and Tables

**Figure 1 ijms-24-12402-f001:**
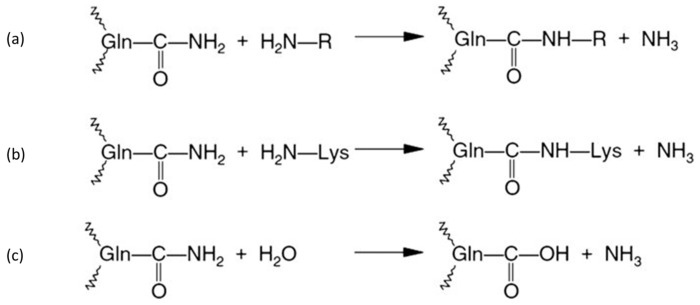
TGM-mediated reactions: (**a**) acyl transfer reaction, (**b**) protein crosslinking reaction, (**c**) deamidation.

**Figure 2 ijms-24-12402-f002:**
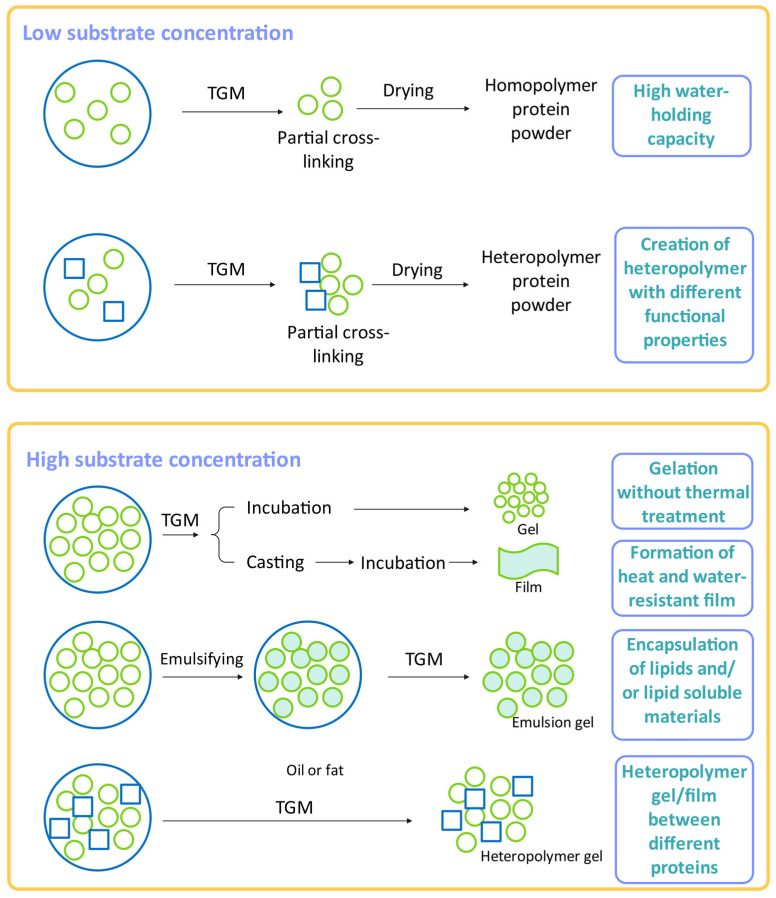
TGM-promoted crosslinks based on low and high substrate concentrations that can produce proteins with new and unique functional properties.

**Figure 4 ijms-24-12402-f004:**
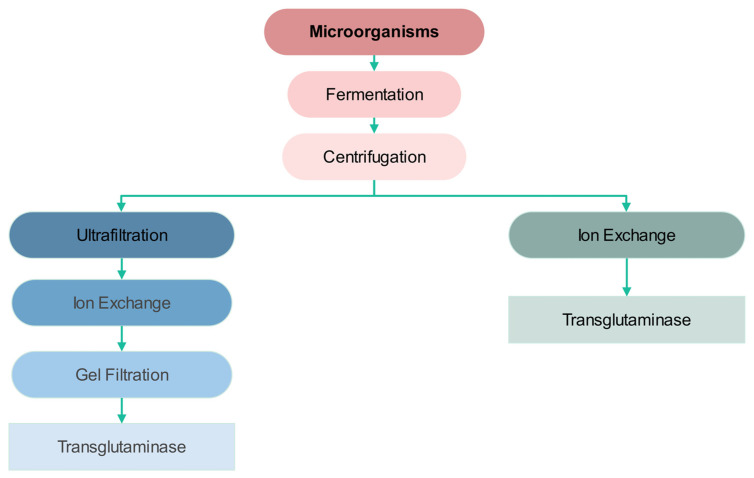
Production of TGM by microorganisms.

**Figure 5 ijms-24-12402-f005:**
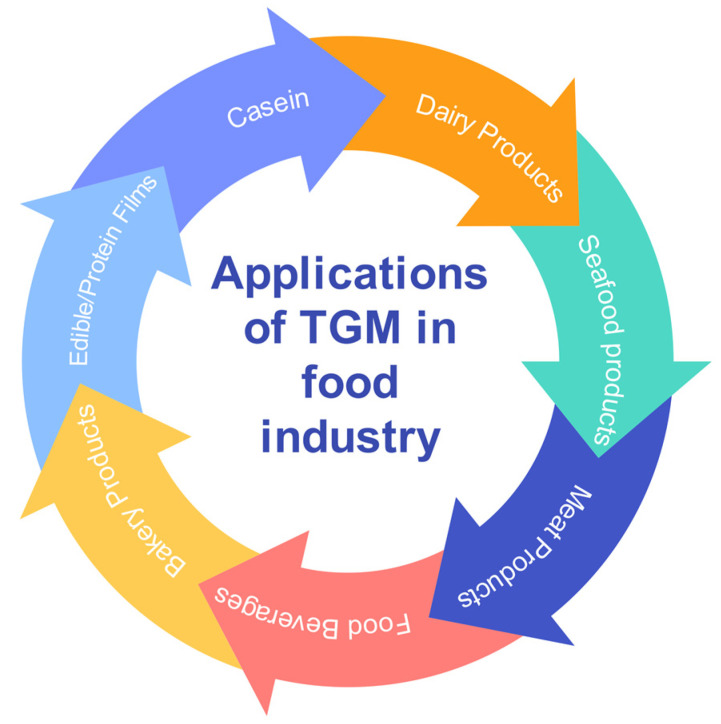
Opportunities for applications of TGM in the food industry.

**Figure 6 ijms-24-12402-f006:**
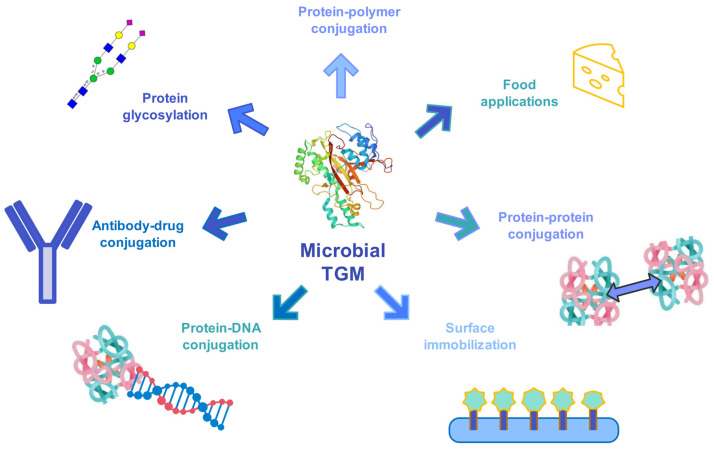
Biotechnological applications of microbial TGM—an overview.

**Figure 7 ijms-24-12402-f007:**
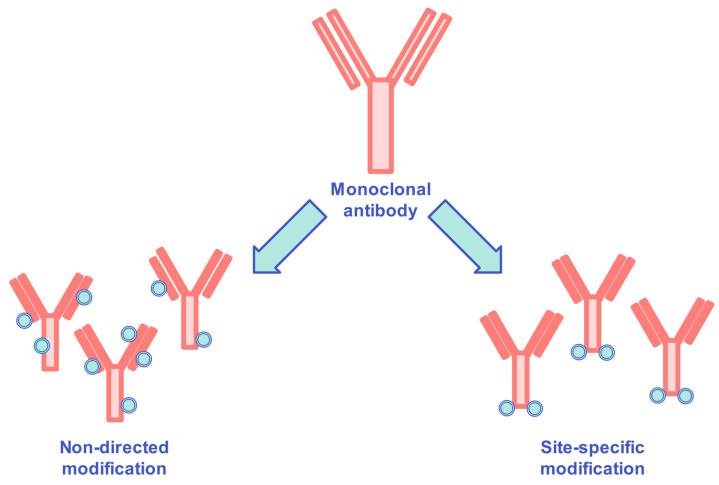
Schematic variations of non-directed modification and site-specific modification of antibody–drug conjugates.

**Figure 8 ijms-24-12402-f008:**
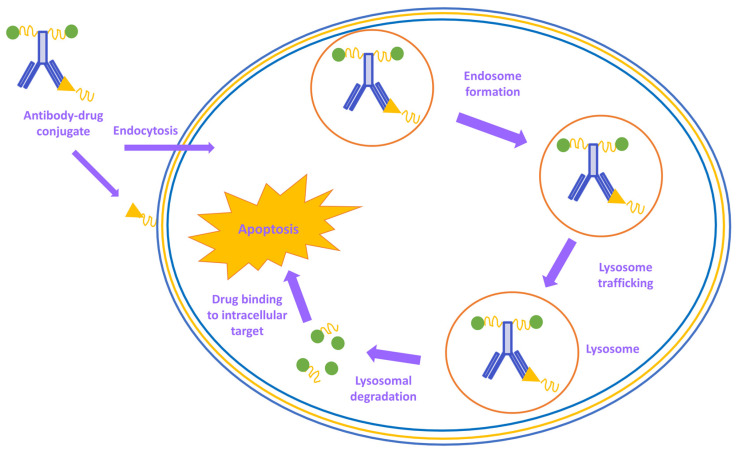
Schematic biological activity mechanism of antibody–drug conjugates.

**Table 1 ijms-24-12402-t001:** Industrial enzymes and their significant applications.

Enzyme	Substrate	Industrial Application
Amylase	Carbohydrate	Detergents, Paper and pulp, Textile, Baking, Starch, Fuel
Laccase	Benzenediol	Textiles, Paper and pulp, Food
Lipase	Fat, oil	Detergents, Oil and fat, Food and baking, Paper and pulp, Fine chemicals
Pectinase	Pectin	Food, Beverages, Textiles
Protease	Protein, polypeptide	Detergents, Food and Leather processing, Water treatment, Animal feeds
Pullulanase	Polysaccharide	Food, Starch
TGM	Protein, amine	Cosmetics, Textiles, Food
Xylase	Xylan	Animal feeds, Baking and food, Paper and pulp

**Table 2 ijms-24-12402-t002:** Properties and differences in TGMs from different sources.

Condition	TGM from Microbial Sources	TGM from Animal Sources
Temperature (°C)	45–55	50–55
pH	5–8	6
Isoelectric point	9	4.5
MW (kDa)	37,800	76,600

**Table 3 ijms-24-12402-t003:** Various microbial strain sources for the isolation of TGM.

Microorganism	Reference
*Bacillus subtilis*	[57,58,59,60]
*Escherichia coli*	[46,47,61]
*Kutzneria albida*	[62,63]
*Physarum polycephalum*	[64,65,66,67]
*Pseudomonas aeruginosa*	[68]
*Sterptoverticilliu mobaraensis*	[69,70,71,72]
*Streptomyces hygroscopicus*	[73,74,75,76,77,78]
*Streptomyces ladakanum*	[79,80]
*Streptomyces libani*	[81]
*Streptomyces nigrescens*	[82]
*Streptomyces platensis*	[80,83,84]
*Streptomyces sioyaensis*	[85]
*Streptoverticillium cinnamoneum*	[86,87,88]

**Table 4 ijms-24-12402-t004:** Reactivity of microbial TGM in relation to different food proteins.

Food Protein	Improved Functional Properties	Reactivity
Egg	Ovalbumin (egg white)	Depending on condition
	Egg yolk protein	Well
Meat	Myoglobin	Depending on condition
	Gelatin	Very well
	Collagen	Well
	Myosin	Very well
	Actin	Does not react
Milk	Casein	Very well
	α-lactalbumin	Depending on condition
	β-lactoglobulin	Depending on condition
	Sodium caseinate	Very well
Soybean	11S Globulin	Very well
	7S globulin	Very well
Wheat	Gliadin	Well
	Glutenin	Well

Some examples of the TGMs used in the food industry are listed below.

## Data Availability

No new data were created or analyzed in this study. Data sharing is not applicable to this article.

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
