# Peer review of "Transglutaminase in Foods and Biotechnology"

_ijms, 2023, doi:10.3390/ijms241512402_

Round 1

Reviewer 1 Report

Manuscript gives very interesting and thorough insight into properties and application of transglutaminases, hence deserve publications after minor revisions.

Remarks:

Title should be changed because it is a bit misleading. Immobilization of TGM is only minor part of review. Also, TGM is not "immobilizing tool", it is object of immobilization process.

Lines 46-49: Rephrase sentence it is a bit confusing, especially because of this "...enzymology for protein modification..." part.

Line 69: "Source" or "origin" is sufficient.

Lines 82-84: Authors should simplify sentence to be more comprehensive. For example: "...the γ-carboxyamide group within glutamine side chain of peptides and..."

Figure 2: Authors should elaborate Fig. 2 more in body of manuscript, especially part about processes at high substrate concentration.

Lines 167-168: "Downstream processes" is more accurate than "mechanism processes".

Lines 174-175: "...the procaryotic TGMs are found in microorganisms, where they are involved in formation of aerial hyphae and sporulation." First part is redundant since in previous paragraph authors described how TGM is produced by bacteria. And second part is misleading because authors talk about procaryotic organisms and then about hyphae, which are characteristic of fungi.

Line 183: If they are result of side-reactions then "by-products" is more appropriate term than "impurities".

Typing errors: line 189 "increses", line 191 "mosty", line 250 "suchs", line 322 "batter".

Lines 198-201: Rephrase sentence. It is pointless to compare conditions of enzyme catalysis and chemical modification of enzyme.

Line 209: Rephrase "...the form of choice considered."

There are several sentences that should be rephrased to provide better understanding by future readers.

Author Response

We thank the reviewer for finding our work valuable for publication.

Remarks:

Title should be changed because it is a bit misleading. Immobilization of TGM is only minor part of review. Also, TGM is not "immobilizing tool", it is object of immobilization process.

The title of our review paper was changed to »Transglutaminase in foods and biotechnology«

Lines 46-49: Rephrase sentence it is a bit confusing, especially because of this "...enzymology for protein modification..." part.

The sentencas was rephrased as suggested by the review:

»Among other approaches, when compared to chemical modifications, applying en-zymes in protein modification displays many advantages, which include high reaction specificities and low side-reactions frequencies, with the lack of need for high pressure and high temperature conditions.« Line 45-48.

Line 69: "Source" or "origin" is sufficient.

The sentence was changed to: “The main differences between microbial TGM and existing TGMs from animal source are presented in Table 2.” Line 68-69.

Lines 82-84: Authors should simplify sentence to be more comprehensive. For example: "...the γ-carboxyamide group within glutamine side chain of peptides and..."

The sentence was rephrased, as suggested by the reviewer: “TGM modifies proteins with amine incorporation and cross-linking, where TGM catalyzes the reaction of the acyl transfer between the γ-carboxyamide group of peptides, which is bound with glutamine residue acyl donors and primary amines receptors of different compounds.” Line 80-83.

Figure 2: Authors should elaborate Fig. 2 more in body of manuscript, especially part about processes at high substrate concentration.

Lines 167-168: "Downstream processes" is more accurate than "mechanism processes".

The sentence was changed to: “However, nowadays, TGM is usually produced by Streptovitricillium mobaraense in fermentation systems, that are followed by the downstream process presented in Figure 4.” Line 164-166

Lines 174-175: "...the procaryotic TGMs are found in microorganisms, where they are involved in formation of aerial hyphae and sporulation." First part is redundant since in previous paragraph authors described how TGM is produced by bacteria. And second part is misleading because authors talk about procaryotic organisms and then about hyphae, which are characteristic of fungi.

The sentence was changed to: “As eucaryotic TGMs are found in phylogenetic taxonomic groups, such as plants, animal and fungi, the procaryotic TGMs are found in microorganisms.” Line 172-173

Line 183: If they are result of side-reactions then "by-products" is more appropriate term than "impurities".

We thank the reviewer for the suggestion, the sentence was corrected to: “For biotechnological technologies, highly pure enzyme TGM is recommended, to sur-pass by-products that can appear during side-reactions.” Line 180-182.

Typing errors: line 189 "increses", line 191 "mosty", line 250 "suchs", line 322 "batter".

All spelling mistakes were corrected throughout the manuscript.

Lines 198-201: Rephrase sentence. It is pointless to compare conditions of enzyme catalysis and chemical modification of enzyme.

The sentence was rephrased as suggested by the reviewer: “Harsh conditions are usually applied during chemical immobilization. If compared to enzymatic catalysis, selective and fast performance in aqueous media is performed under mild process conditions, which has many advantages.” Line 196-199.

Line 209: Rephrase "...the form of choice considered."

The sentence was rephrased: “For larger-scale applications, immobilized enzymes are the considered subject of choice.” Line 206-207.

Reviewer 2 Report

The paper is a thorough and comprehensive review on transglutaminases. It describes their sources, properties, applications in different industries, such as food-related, biotechnology and cosmetics. In my opinion this is a valuable work that is of great interest to be published.

Minor revision of the paper is suggested by the reviewer, while there are few repetitions, such as in chapter 3, lines 132-133, and lines 147-148. These facts were already mentioned earlier, so it should be rephrased.

Page 1, lines 30-32 Please rewrite the sentence starting with: Rising concerns... The link of this sentence to the rest of the text is somehow missing.

Also, page 3, lines 97-99 Please elaborate and clarify this sentence.

Author Response

The paper is a thorough and comprehensive review on transglutaminases. It describes their sources, properties, applications in different industries, such as food-related, biotechnology and cosmetics. In my opinion this is a valuable work that is of great interest to be published.

We thank the reviewer for finding our work valuable for publication.

Minor revision of the paper is suggested by the reviewer, while there are few repetitions, such as in chapter 3, lines 132-133, and lines 147-148. These facts were already mentioned earlier, so it should be rephrased.

All sentences were rephrased, as suggested by the reviewer. Line 130-137.

Page 1, lines 30-32 Please rewrite the sentence starting with: Rising concerns... The link of this sentence to the rest of the text is somehow missing.

The sentence was rephrased: »Raising concerns of global food shortages and population growth, as well as the use of advanced food proteins, is increasing constantly in recent years.” Line 29-31.

Also, page 3, lines 97-99 Please elaborate and clarify this sentence.

The sentence was rephrased: »The TGM cross-linking reaction occurs before the acyl transfer and deamidation reactions in food systems, which results in the formation of glutamyl lysine isopeptides and polymers with high molecular weight. Consequently, it changes the functional properties of the proteins, resulting in improved rheology and other quality properties for various food products (Figure 2) [26].« Line 93-97.
